# Effects of Physical Activity on Cognition, Behavior, and Motor Skills in Youth with Autism Spectrum Disorder: A Systematic Review of Intervention Studies

**DOI:** 10.3390/bs14040330

**Published:** 2024-04-15

**Authors:** Sara Suárez-Manzano, Alberto Ruiz-Ariza, Nuno Eduardo Marques de Loureiro, Emilio J. Martínez-López

**Affiliations:** 1Research Group HUM-943, Faculty of Educational Sciences, University of Jaén, 23071 Jaén, Spain; ssuarez@ujaen.es (S.S.-M.); emilioml@ujaen.es (E.J.M.-L.); 2Laboratorio de Actividade Física e Saúde, Escola Superior de Educação, Instituto Politécnico de Beja, 7800-295 Beja, Portugal; nloureiro@ipbeja.pt

**Keywords:** academic performance, adolescent, ASD, cognitive performance, physical exercise, stereotypic and repetitive movements

## Abstract

The aim of this paper was to analyze the acute and chronic effects of physical activity (PA) on cognition, behavior, and motor skill in youth with autism spectrum disorder (ASD), taking into account potential confounders. In addition, it was intended to elaborate a guide of educational applications with strategies for PA use. Studies were identified in four databases from January 2010 to June 2023. A total of 19 interventional studies met the inclusion criteria. PA programs ranged from two weeks to one year in duration, with a frequency of one to five sessions per week. More than 58% of the studies showed positive effects of PA on cognition, and 45.5% on behavior and motor skill. Moderate–vigorous PA for 15–30 min has shown acute effects on cognition, general behavior, and stereotypic/repetitive behaviors in youth with ASD. A total of 9 out of 14 studies showed chronic effects on general behavior and stereotypic behaviors, and only 6 on motor skills.

## 1. Introduction

Autism spectrum disorder (ASD) is characterized by abnormalities in social interactions, impairments in language and communication, restrictive or repetitive interests, stereotypic behaviors, perceptual difficulties, sensory abnormalities, and intellectual delay. ASD includes autistic disorder, Asperger syndrome, and pervasive developmental disorder not otherwise specified [1]. Likewise, symptoms of attention deficit hyperactivity disorder are common in young people on the autism spectrum without intellectual disabilities [2]. The origin of this pathology may be due to genetic factors, environmental factors, or the combination of both [3]. The global prevalence rate of ASD in young people of ≤18 years of age is around 0.62–0.70% and shows a male-to-female ratio of 3–4:1, with no differences by age [4,5].

At a general level, cognitively, these children can present everything from exceptional abilities in specific areas to significant challenges in learning and attention. In terms of behavior, it is common to observe repetitive patterns, resistance to changes in routine, and, in some cases, self-harming behaviors. Motor skills may also be compromised, manifesting in limited coordination and fine motor skills. In terms of communication and social interaction, autistic children may have difficulties in understanding and using verbal and non-verbal language, which affects their ability to interact effectively with others, establish relationships, and understand social norms, contributing to the complexity of the disorder [1].

The main conditions of ASD in schoolchildren are related to three elements that are considered essential for the learning process. First, there is cognition, which includes cognitive performance—including number of abilities, automatic responses, resisting distraction or interference, working memory, dual tasking, planning, monitoring, and verbal and design fluency [6]—and academic performance, measured through standardized performance tests or through marks in different school subjects [7,8]. The second element is behavior and emotional regulation, including social/communication and stereotypic and self-injurious behaviors [9,10]. The third element is motor skill, including stereotypic and repetitive movements, such as shaking one’s hands or oscillating one’s body [11].

Scientific evidence has shown that, to treat ASD in schoolchildren, it is necessary to act based on two approaches. The first is medication, such as the following: Methylphenidate, Atomoxetine, and glutamatergic medication are the pharmacological treatments most used to treat this kind of disorder, because ASD is mainly related to high levels of serotonin and low levels of dopamine in the brain [12], low activity of synaptogenesis, and an imbalance between excitatory and inhibitory currents [13]. The second approach is psychosocial treatments, incorporating video technology and telehealth during the last decade to coach parents [14]. Other studies indicate that alternative treatments, such as equine-assisted activities and therapies [15], social–behavioral interventions [16], and mindfulness programs [17], provide improvements such as greater social interaction, decreased problem behaviors, and improved inhibitory abilities, respectively.

In recent years, physical activity (PA) has appeared as a complement to previous treatments. Numerous studies affirm the positive effect of PA on executive functions, behavior, and motor skills. The interventions are very varied, such as swimming, running, walking, horseback riding, exergaming, or yoga, among others. In general, the intervention programs are between 8 and 14 weeks long, with interventions of between 2 and 4 days/week, lasting between 45 and 60 min [18,19]. Sansi et al. (2020) [19] found that children with ASD improved their motor skills, social skills, and attitudes after 50 min of PA twice a week for 12 weeks. On the other hand, Nakutin and Gutierrez (2019) [20] observed improvements in academic engagement in a sample of three children with ASD, following a four-week-long program (12 min jogging/session, two session/week). These effects could be explained by an increase in the level of noradrenaline and dopamine in the encephalon [12] and a biological response of the adaptation of brain functions to the stimulus generated by physical exercise [21]. This adaptive response may enhance cognitive functions by improving attention, memory, and executive functions, which are critical for learning and daily activities. In fact, exercise practice and relaxation decrease cortisol levels, promoting the decrease in anxiety and stress [22], the improvement of brain-derived neurotrophic factor (BDNF) and insulin-like growth factor-1 [23], and the increase in executive functions, with performance more consistently benefitting inhibition tasks compared to dual-task coordination, shifting tasks, or combined tasks [24]. Ohara et al. (2020) [25] observed that children with ASD with weaker motor skills have higher social skill deficits. The parents of ASD schoolchildren are aware of the benefits of PA for their health and that of their children [26]. In parallel, a recent systematic review has stated that movement/sports training can help to improve fundamental motor skills in young adults with ASD (between 19 and 30 years old) [27].

Furthermore, the benefits of PA in youth with ASD have been shown to be broadened according to practice time. The acute effect of PA is associated with an improvement in cognition and a reduction in stereotypic behaviors [28,29,30]. However, the chronic effect of PA over two weeks, in addition to producing cognitive improvements [31], induces behavior improvements [32], promotes the learning of motor skills [19,33,34] and sleep efficiency [35], and decreases anxiety and stress levels [22], as well as moodiness [35].

Despite the research noted above, the acute and chronic effects of PA on the levels of cognition, behavior, and motor skill are still inconclusive. Some research suggests that the results come from studies in which the multifactorial nature of PA is not differentiated [8,36]. Children and adolescents take part in sports [37], PE classes, and extracurricular PA [38], which are activities that most studies do not distinguish in their results. In addition, the known reviews analyzing the effect of PA on cognition and/or behavior in youth with ASD include a small number of longitudinal studies, where none of them explicitly differentiate between acute and chronic effects, and they do not report the effects of the exercise [39].

The main question for this study was the following: “Does systematic physical activity improve cognition, behavior, and motor skill in youth with ASD?” The present review is focused on children and adolescents because, during this period, the young maintain a high degree of brain plasticity necessary for learning, and it is the best for inducing young people to participate in systematic PA [8]. Understanding the effects of PA on cognition, behavior, and motor skill in youth with ASD could help to raise awareness about the importance of PA in this population, serve as a complement to traditional treatments based on medication, and make decisions about the level of PA integration in education systems.

In this systematic review, we analyzed the results of interventional studies examining the acute and chronic effects of PA on cognition, behavior, and motor skill in children and adolescents with ASD. Additionally, this paper also proposes the following: (1) To review the potential mediators and moderators (i.e., socioeconomic variables, fatness, or gender) that could influence the relationship between PA and cognition, behavior, and motor skill. (2) To elaborate a didactic application for PA use and its possible benefits on cognition, behavior, and motor skill associated with its practice.

## 2. Methods

This study was designed following the structure and recommendation of PRISMA guidelines for reports and studies [40]. In accordance with Cochrane’s indications for systematic reviews of interventions, Table 1 shows the databases, search strategies and limits, and filtered papers to allow replication of the study [41]. Moreover, we consulted the Newcastle–Ottawa approach for observation studies, and we registered the systematic review in PROSPERO (CRD42016051631).

### 2.1. Search Limits

An extensive investigation was conducted across four major literature databases (PubMed, Web of Science, SCOPUS, and Medline) spanning from January 2010 to June 2023. Furthermore, the bibliographies of the chosen articles were scrutinized. Additionally, the reference lists of the selected papers were reviewed. Key search term categories were identified and used in different combinations, as follows:(1)Physical activity (physical fitness, cardiovascular fitness, physical activity, physical education, fitness, exercise, physical exercise, healthy exercise, aerobic exercise, resistance exercise, and anaerobic exercise).(2)Autism spectrum disorder (ADD, Asperger, autism, and autistic).(3)Cognition (cognition, academic, cognitive, executive function, memory, attention, creativity, perception, and behavior)(4)Children and adolescents (adolescent, teenagers, children, childhood, school-age youth, and student).

### 2.2. Selection Criteria

The relevant papers selected for inclusion in the review were checked against the following criteria: (1) the study was a full report published in a peer-reviewed journal; (2) the study population was youth with ASD; (3) the study included papers written in English, French, or Spanish, with a population of students between 3 and 21 years; and (4) the study used an interventional study design.

### 2.3. Data Extraction and Reliability

The search process was carried out by three independent reviewers (S.S.M., A.R.A., and E.J.M.L.). They read every title and all of the abstracts, and a consensus meeting was held to resolve any differences between them. Information on the authors, title, aim, sample size, age, country, design, PA measurement, ASD measurement, confounders, and main results/conclusions was extracted from each study. The results of the most recent reviews were summarized first, and then the studies that were potentially relevant for the selected topics were screened for retrieval.

### 2.4. Quality Assessment and Level of Evidence

A quality assessment was carried out on the basis of other standardized assessment lists [8] and on our selection criteria (Table 2). The list included the following six items: (A) the study was a full report published in a peer-reviewed journal; (B) the study population had autism spectrum disorder (ASD); (C) the selected physical activity, cognition, and behavior outcomes were clearly described; (D) the population was made up of Pre-school, Primary, and High School youths between 6 and 18 years of age; (E) the study had an interventional design; and (F) the data were adjusted for confounders. Each item was rated as “2” (fully reported), “1” (moderately reported), or “0” (not reported or unclear). For all studies, a total quality score was calculated by counting up the number of positive items (a total score of between 0 and 12). Three levels of evidence were constructed. Studies were defined as high quality (HQ) if they had a total score of nine or higher. A total score of five to eight was defined as medium quality (MQ), and a score of less than five was defined as low quality (LQ). After the analysis, all of the included articles obtained an HQ score.

## 3. Results

### 3.1. General Findings

The flow of search results through the systematic review process is shown in Figure 1, resulting in an initial search of 842 papers. After the removal of duplicates (296), a total of 546 papers were retrieved. In the following step, a total of 398 papers were excluded by population, age, or language (267); or by design or variables (131) not in line with our inclusion criteria. Thus, 148 potential studies were reviewed for inclusion criteria. Finally, 19 studies were included in the systematic review. A detailed analysis of these studies showed that all 19 were intervention studies (100%), of which 5 (21%) were acute exercise intervention [28,30,45,51] and 14 (79%) were chronic exercise intervention [20,21,22,31,32,33,35,38,43,44,46,47,48,49,50]. Detailed information about all of the studies is presented in Table 3.

### 3.2. Acute Effect of Physical Activity on Cognition, Behavior, and Motor Skill

Only one of five studies analyzed the acute effect of PA on cognition-controlled exercise intensity through heart rate [28]. These five studies used short-term interventions at moderate–vigorous intensity [28,30,42,45,51]. Anderson-Hanley et al. (2011) [42] studied a stimulus based on 20 min of exergames in 24 adolescents (10–18 years) with ASD. The experimental group improved in repetitive behaviors (Gilliam Autism Rating Scale, 2nd edition) and executive function, measured with the digit span forward and backward test, with no differences in the results of the Color Trails Test or Stroop Test. Similarly, Golden et al. (2022) [30] analyzed the effect of 20 min of active video gaming or brisk walking on response time and accuracy. Both experimental groups improved compared to the control group. Liu and Hamilton (2013) [28] found a similar effect in children with ASD. They found that after four different sessions of PA—15 min/day at 100–140 beats/minute—similar improvements were produced in stereotypic behaviors and in task-engaged behavior, while finding no differences related to the confounders age, gender, or disorder type. Contradictory results were observed compared to those observed by Ludyga et al. (2023) [51], who analyzed the acute effect of 20 min of moderately intense cycling on face recognition task (reaction time, correct responses, and accuracy rates). The results showed a greater increase in reaction time in the exercise group compared to the control group. Finally, in the study of Obrusnikova et al. (2012) [45], four children aged 9–11 years took part in two afternoon sessions. In the first, they performed at four fitness stations and two skill stations at moderate–vigorous intensity. During the second session, each child was accompanied by a dog. In both cases, benefits were observed in learning, behavior, and moderate-to-vigorous PA (MVPA). However, the gains in academic learning time–PE and task behavior were greater when they conducted the session accompanied by dogs.

### 3.3. Chronic Effect of Physical Activity on Cognition, Behavior, and Motor Skill

A total of 14 longitudinal studies with intervention analyzed the chronic effect of PA on cognition. Brand et al. (2015) [35], after several afternoon PA sessions (60 min/day × 3 sessions/week × 3 weeks), observed improvements in sleep efficiency, motor skills, mood, and behavior. Zhang et al. (2020) [47] found that preschoolers with ASD aged between 3 and 6 years who performed basketball training—40 min/day × 5 sessions/weeks × 12-week—showed improvements in their working memory. Likewise, it was observed that 12 weeks of intervention with basketball sessions—45 min/day × 2 sessions/week—in ASD children between 8 and 12 years improved their inhibitory control [31].

On the other hand, Hillier et al. (2011) [22] observed that physical exercise and relaxation for 75 min per day, once a week for 8 weeks, caused a decrease in cortisol levels, reducing anxiety and stress levels in adolescents with ASD. A motor skill instruction program based on the practical application of many skills—4 h/day × 5 sessions/week × 8 weeks—induced improvements in the behavior of children between four and six years of age, especially in all three motor outcomes: locomotor, object control, and gross quotient [33]. Nicholson et al. (2011) [32] found that children in a PA program conducted just before the beginning of the school day—17 min/day × 3 sessions/week × 2 weeks—showed a more active attitude to learning and obtained large effect sizes for academic engagement. These results were similar to those obtained by Nakutin and Gutierrez (2019) [20]—12 min/session jogging × 2 session/week × 7 weeks. Rafiei et al. (2021) [48] observed improvements in executive function (Wisconsin Card Sorting Test) in both experimental groups (Kinect or Sports, Play, and Active Recreation for Kids (SPARK))—35 min/session × 3 session/week × 8 weeks—after school. These results were similar to those obtained by Haghighi et al. (2022) [50]—40–60 min/session of combined physical training × 3 session/week × 8 weeks—using the Gilliam Autism Rating Scale, second edition test (GARS-2), and Marzouki et al. (2022) [51], who also observed improvements in emotional functioning after 8 weeks of intervention with technical or game-based aquatic activity programs (50 min/session × 2 session/week), with respect to the control group.

Oriel et al. (2011) [43] obtained improvements in correct responses after a running intervention before the school day, consisting of 15 min/day × 5 sessions/week × 3 weeks; however, they did not find improvements in general behaviors or stereotypic behaviors. Coinciding with Andy (2020) [45] and Zhang et al. (2020) [47], improvements in regulation behavior obtained after 12 weeks of intervention were also observed.

In children who are 6–12 years old, it has been proven that executive functions and motor skill improved after each PA session of 70 min carried out once a week for 12 weeks [34]. After 12 follow-up weeks after the intervention, it was also proved that the efficiency was maintained for at least 2 weeks. Finally, Vander and Sprong (2011) [48] revealed that children who were 3–6 years old who performed PA before the school day (15 min of running/day × 3 sessions/week × 3 weeks) increased their academic performance, answering a greater number of questions and obtaining a greater number of correct answers during the school day. However, they did not significantly improve in general behavior or stereotypic behaviors.

## 4. Discussion

This systematic review has analyzed the acute and chronic effects of PA on cognition, behavior, and motor skills in children and adolescents with ASD. Among the 19 studies reviewed, 57.9% showed positive effects of PA on cognition and 45.5% on behavior and motor skills. The five studies with acute intervention showed that PA sessions employing mainly moderate–vigorous intensity exercises for 15–30 min have a positive acute effect on cognition, general behaviors, and stereotypic/repetitive behaviors in youth with ASD. Nevertheless, only two studies showed an improvement in executive functions (digits backwards and Wisconsin Card Sorting Test). The 10 studies that analyzed the chronic effect of PA employed mainly aerobic exercise before the school day or activities to improve motor skills during the afternoon. The duration ranged between two weeks and one year, and the exercise was based on MVPA (from one to five sessions/week). The session time was between fifteen minutes and four hours; moreover, the short-term sessions were carried out at moderate–vigorous intensity, and the long-term sessions at moderate intensity. Nine studies showed a positive effect of PA on general behaviors and stereotypic behaviors, and five studies reflected an improvement in motor skills.

A general analysis of the selected studies has revealed that PA improves cognition, behavior [48], and motor skills [50] in youth with ASD. However, this study has shown that PA benefits are different according to the intervention time, and their short- and long-term effects have different features that deserve to be analyzed separately. During the last decade, studies have proliferated, demonstrating the need to use PA as a means to educate in leisure time, especially in physical education (PE) [36,47]. In Table 4, we present didactic strategies targeted to manage challenging behaviors and to improve cognition through PA practice in youth with ASD. Both currently represent a disability that is often experienced by teachers working with special education students. Martínez-López et al. (2021) [52], in a systematic review carried out on typically developing schoolchildren, analyzed the methods to include PA in the school context. If we compare the results, we can find active breaks and PAAC and PAAL methods, which can be two useful strategies for schoolchildren diagnosed with ASD, since the characteristics of the sessions coincide with those presented by the authors (Table 4).

The analysis of the acute effects of PA has shown that isolated sessions, using exercise at moderate–vigorous intensity for 15–20 min, are enough to improve cognition, general behaviors, and stereotypic/repetitive behaviors in youth with ASD [28,30,42,44,51]. On the other hand, the analysis of the chronic effects of PA has shown improvements mainly in academic performance [20], cognition [19,31,44,47], and motor skills [38,45,47,50] in youth with ASD. These results are similar to those obtained by Tan (2013) [53] in 12 boys with ASD aged two to six years. After performing eight sessions of tri-cycling for 15 min at a moderate–vigorous intensity, it was found that the PA group had an increasingly longer attention span compared with the non-PA group. The results referring to improvements in motor skills are in line with the results obtained by Pan (2011) [54], who observed an increase in physical fitness and aquatic skills after an aquatic program of 60 min twice a week over 14 weeks, at moderate intensity. Pan et al. (2017) [38] explained that, after an intervention with exergames over 12 weeks, the improvements obtained in motor skills and executive function remained during a period of at least equal to the intervention time in youth with ASD. These improvements in motor skills also imply improvements in social communicative skills [25]. A recent systematic review has also demonstrated that horseback riding and martial arts interventions may produce the greatest results in improvements to numerous behavioral outcomes, including stereotypic behaviors, social–emotional functioning, cognition, and attention [39]. The above suggests setting up a PA promotion policy within schools, too.

Some factors would explain the acute and chronic effects of PA on the cognition, behavior, and motor skills of children and adolescents. Performing a single PA session would increase brain function, for example [12]. The above is due to the activation of synaptogenesis, an immediate effect of neuronal functions and brain activation that favors cognitive performance [55]. However, at a chronic level, in addition to cognitive and behavioral benefits, improvements in motor skills and sleep quality in youth with ASD were obtained. All of this could be because the MVPA activates the PGC-1α–Errα transcription complex, which stimulates the generation of the FNDC5 gene in the hypothalamus, and, in turn, the FNDC5 gene stimulates the BDNF gene, the master regulator of cell survival, differentiation, and plasticity in the brain, which improves executive functions [24]. Secondly, exercise and its effect on relaxation decrease cortisol levels, promoting a reduction in anxiety and stress [22], improving the quality of sleep and mood, and favoring the improvement of the general and stereotypic behaviors of youth with ASD [22,56]. Finally, land-based PA intervention [57] and aquatic programs [49] improve motor and behavior skills in youth with ASD because of improvements in hypotonia and looseness of muscles, strength, balance, and agility skills [48].

In addition to the direct effects of PA on cognition, an inverse relationship was found between PA time and minutes isolated. That is, as youth with ASD progressed in the PA program, they exhibited decreased loneliness [33]. On the other hand, no differences in the effect on cognitive performance have been found based on age, gender, or subtype of ASD [28]. In the other studies, covariates were not included in the analyses, perhaps due to the small sample size [n = 4] [32,45] or because they were pilot studies [22,35]. Most of the studies analyzed only the direct effects of PA on cognition, behavior, and motor skills, thus obtaining results that are probably biased. In addition, most of the studies carried out in the ASD population do not consider the other variables usually used in studies with a school-age population, such as BMI [58], commuting time to school [8], or the possible presence of the phenomenon of ‘intellectual giftedness and ASD’, which can be detected using systematic identification measures [59]. Because it is known that youth with ASD have lower levels of moderate PA during the school day and perform less daily PA than their typically developing peers [34], it would be interesting to know if the previous variables influence this population, in addition to other variables such as sleep quality and parental influence, in a similar way.

### Limitations and Future Directions

This systematic review shows the inaccuracy of some applied PA programs as the main weakness. Some studies do not take into account the PA that participants perform throughout their daily lives, which could bias the results. These limitations make it difficult to know the effect of the isolated program. Additionally, some studies have a sample size of less than 10 participants. Despite the above, it is the first time that a systematic review classifies PA into acute and chronic effects and studies these variables in schoolchildren diagnosed with ASD. In this line of research, for future research, follow-up is also recommended in order to analyze the duration of the effects. In addition, most of the studies carried out in the ASD population do not take into account the other variables usually used in studies with a school-age population, such as age, sex, and BMI [58]. In a similar way, it is recommended to monitor the participants’ daily PA with the use of accelerometry or smart bracelet, because it is known that youth with ASD carry out lower levels of moderate PA during the school day and perform less daily PA than their typically developing peers [34]. Moreover, would be interesting to know if the previous variables influence this population, in addition to other variables such as sleep quality and parental influence.

## 5. Conclusions

In the present review, we found a total of 20 longitudinal studies with intervention with high-quality criteria. Overall, 64% of the studies showed a positive effect of PA on cognition, and 46% on behavior and motor skills of youth with ASD. The application of PA sessions that mainly employ moderate–vigorous intensity exercises for 15–30 min has shown an acute positive effect on cognition, general behaviors, and stereotypic/repetitive behaviors in youth with ASD in all cases, and only sometimes on executive functions. A total of 10 of the 15 studies that analyzed the chronic effect of PA—mainly using aerobic exercise before the school day or activities to improve motor skills during the afternoon—showed a positive effect on general behaviors and stereotypic behaviors, and only 6 studies reflected an improvement in motor skills. No study revealed a negative association. Based on the scientific evidence of the articles analyzed, practical applications are proposed and presented in Table 4.

The articles analyzed here do not report significant differences in the effects of PA on the analyzed variables concerning the age of the participants. However, it is a determining factor for the design of educational applications. They must be adapted to the developmental stage of the child or adolescent, due to the variability in individual responses to exercise reflected in studies conducted on youth without ASD [52]. Similarly, no different effects have been found according to the sex of the participants. It is notable that many studies only involved boys, and, in most, there was a higher number of boys than girls, which is possibly due to a higher prevalence of ASD in boys. However, studies on the non-ASD population state that girls tend to engage in less PA, therefore, the intervention’s effect is greater, based on the dose–response phenomenon [8].

Although this article highlights the benefits of PA, it suggests the need for further investigation into how additional variables, such as BMI, travel time to school, and the coexistence of intellectual giftedness and ASD, can influence the observed benefits of PA in this population. Moreover, researching the long-term effects of different types of PA and their impact on social and communicative skills, sleep quality, and overall well-being would be beneficial. More research is needed to justify the acute and chronic differential effects on cognition, behavior, and motor skills. The results presented here must be considered with caution because most of the studies used small samples and had a limited inclusion of confounders.

## Figures and Tables

**Figure 1 behavsci-14-00330-f001:**
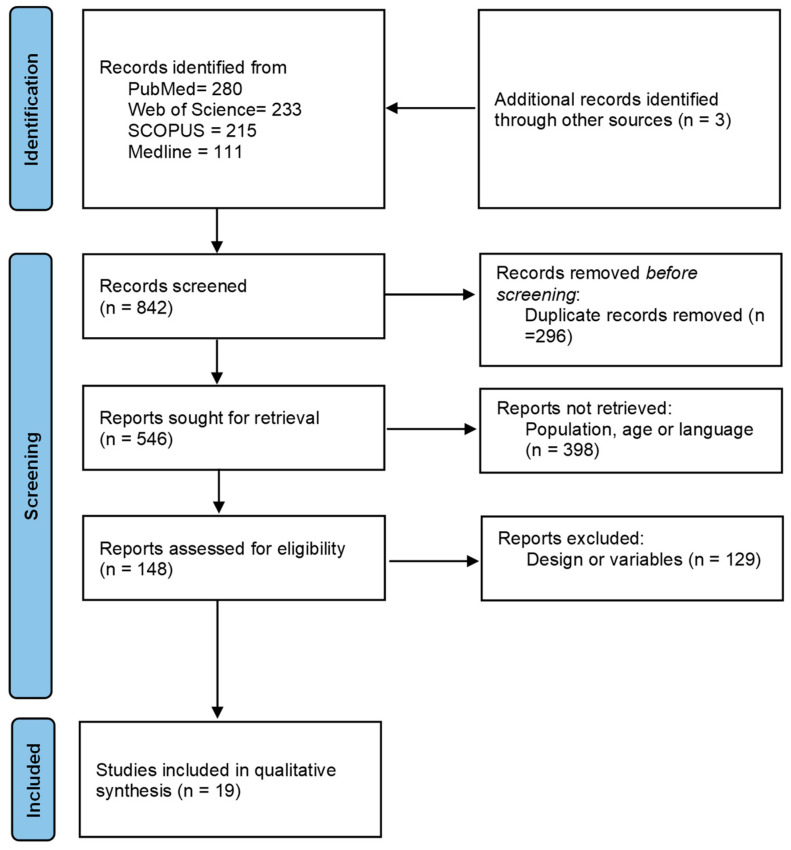
Flowchart based on PRISMA.

**Table 1 behavsci-14-00330-t001:** Search strategy in databases.

Database	Search Strategy	Limits	Filter
PubMed	(“physical fitness” [Title/Abstract] OR “physical activity” [Title/Abstract] OR “physical education” [Title/Abstract] OR “fitness” [Title/Abstract] OR “exercise” [Title/Abstract] OR “physical exercise” [Title/Abstract] OR “acute exercise” [Title/Abstract] OR “chronic exercise” [Title/Abstract] OR “healthy exercise” [Title/Abstract] OR “aerobic exercise” [Title/Abstract] OR “resistance exercise” [Title/Abstract] OR “anaerobic exercise” [Title/Abstract]) AND	Publication date from 1 January 2010 to 1 June 2023	283 items filtered
Web of Science	(“cognition” [Title/Abstract] OR “academic” [Title/Abstract] OR “cognitive” [Title/Abstract] OR “executive function” [Title/Abstract] OR “memory” [Title/Abstract] OR “attention” [Title/Abstract] OR “creativity” [Title/Abstract] OR “perception” [Title/Abstract] OR “behavior” [Title/Abstract]) AND	Species: Humans	233 items filtered
SCOPUS	(“ADD” [Title/Abstract] OR “Asperger” [Title/Abstract] OR “autism” [Title/Abstract] OR “autistic” [Title/Abstract]) AND	Age: 6–12 and 13–18	215 items filtered
Medline	(“children” [Title/Abstract] OR “childhood” [Title/Abstract] OR “school-age youth” [Title/Abstract] OR “adolescent” [Title/Abstract] OR “teenagers” [Title/Abstract] OR “student” [Title/Abstract] OR “school” [Title/Abstract]] OR “high school” [Title/Abstract]))	Language: English, French, and Spanish	111 items filtered

**Table 2 behavsci-14-00330-t002:** List of included studies with quality scores.

Authors and Variables	A	B	C	D	E	F	Total Score	Quality Level
Anderson-Hanley et al. (2011) [42]. Exergaming, behaviors, cognition, and ASD	2	2	2	1	2	0	9	HQ
Hillier et al. (2011) [22]. Exercise, relaxation, and ASD	2	2	2	1	2	0	9	HQ
Nicholson et al. (2011) [32]. Physical activity, academic engagement, and ASD	2	2	2	1	2	0	9	HQ
Oriel et al. (2011) [43]. Aerobic exercise, academic engagement, and ASD	2	2	2	1	2	0	9	HQ
Vander and Sprong (2011) [44]. Aerobic exercise, academic engagement, and ASD	2	2	2	1	2	0	9	HQ
Obrusnikova et al. (2012) [45]. Dog in physical activity and ASD	2	2	2	1	2	0	9	HQ
Liu and Hamilton (2013) [28]. Physical activity, stereotypic behaviors, and ASD	2	2	2	2	2	2	12	HQ
Brand et al. (2015) [35]. Aerobic exercise, sleep, motor skills, and ASD	2	2	2	1	2	0	9	HQ
Ketcheson et al. (2016) [33]. Early motor skill intervention, motor skills, physical activity, socialization, and ASD	2	2	2	1	2	2	11	HQ
Pan et al. (2017) [38]. Physical activity, physical and cognitive outcomes, and ASD	2	2	2	1	2	0	9	HQ
Nakutin and Gutierrez (2019) [20]. Jogging for academic engagement	2	2	2	1	2	0	9	HQ
Tse et al. (2019) [31]. Basketball and inhibitory control	2	2	2	1	2	0	9	HQ
Andy (2020) [46]. Moderate to vigorous jogging and emotion regulation	2	2	2	1	2	0	9	HQ
Zhang et al. (2020) [47]. Mini-basketball, working memory, and regulation	2	2	2	1	2	0	9	HQ
Rafiei et al. (2021) [48]. SPARK and Kinect and executive function	2	2	2	1	2	0	9	HQ
Golden et al. (2022) [30]. Active video gaming, response time, and accuracy	2	2	2	1	2	0	9	HQ
Marzouki et al. (2022) [49]. Combined physical training	2	2	2	1	2	0	9	HQ
Haghighi et al. (2022) [50]. Aquatic training, stereotypic behaviors, communication, and social interaction	2	2	2	1	2	0	9	HQ
Ludyga et al. (2023) [51]. Aerobic exercise and face recognition task	2	2	2	1	2	0	9	HQ

Note: High quality (HQ) = 9–12. AP = Academic performance. CP = Cognitive performance. PA = Physical activity. (A) The study was a full report published in a peer-reviewed journal. (B) The study population had autism spectrum disorder (ASD). (C) The selected physical activity, cognition, and behavior outcomes were clearly described. (D) The population was made up of Pre-school, Primary, and High School youths between 6 and 18 years of age. (E) The study had an interventional design. (F) The data were adjusted for confounders.

**Table 3 behavsci-14-00330-t003:** Characteristics of the analyzed studies (N = 20).

Author	Study Design/Intervention Acute or Chronic Confounders/Duration/Went	Sample/Age (Years)/Country	Groups/Physical Activity Measures	Cognition Measures	Results
Anderson-Hanley et al. (2011) [42]	Interventional, cross-over. Randomly/Acute/Diagnosis with DSM-IV and Gilliam Autism Rating Scale, 2nd edition/2 weeks (2 sessions)/During the summer.	24 adolescent (4 girls)/10–18/14.8 ± 2.7/USA 10 boys and adolescent/8–21/13.2 ± 3.8/USA	Pilot I: 2 groups: EG (n = 12): Exergame 20 min CG (n = 12): Watched a 20 min video Pilot II: 2 groups: EG (n = 10): Exergaming “cybercycle”	Gilliam Autism Rating Scale, 2nd edition: Measurable behaviors and associated features of autism CCTT: Aspects of executive function, including task switching Stroop: Executive function, including inhibition of response Digit span forward and backward: Executive function captured in the backward performance	Repetitive behaviors significantly decreased, while performance on digits backwards improved following the exergaming conditions compared with the control condition.
Hillier et al. (2011) [22]	Interventional, cross-over. Randomly/Chronic/Diagnosis with DSM-IV. Time (each session)/8-week. One session a week/in the early evening (18:15–19:30 h).	18 Adolescents (2 girls)/13–27/17.1/USA	1 group: EG: 75 min physical exercise (aerobic, muscle strengthening, and bone strengthening) Cortisol level 2, 4, 6, and 8	STAI: Self-report anxiety measure before and after each session	A significant reduction in cortisol at the end of the sessions compared with the beginning was observed. This was evaluated using a self-report anxiety measure. While the decreases in stress indicators were not maintained over time, the findings emphasize the potential of exercise and relaxation to improve stress symptoms.
Nicholson et al. (2011) [32]	Interventional, cross-over. Randomly/Chronic/Diagnosis of ASD/2 weeks. Three times per week. Before the classroom.	4 children (all boys)/9/9/USA	1 group: EG: ASD. 12 min jogging intervention + 5 min cool-down	BOSS: On task or academically engaged behavior	Large effect sizes for academic engagement time for all four students. They were more active after the intervention.
Oriel et al. (2011) [43]	Interventional, cross-over. Randomly. Contrabalanced/Chronic/Diagnosis of ASD/3 weeks + 3 weeks. Five times per week. Before the classroom.	9 children (2 girls)/3–6/5.2/USA	2 groups: EG (n = 9): 15 min of running/jogging followed by a classroom task CG (n = 9): Classroom task that was not preceded by aerobic exercise	AP: Correct academic responses, incorrect academic responses, stereotypic behaviors, and on-task behavior	Statistically significant improvements were found in correct responses following exercise (*p* < 0.05). No significant differences were found for on-task behavior or stereotypic behaviors.
Vander and Sprong (2011) [44]	Interventional, cross-over. Four classes were contrabalanced. Randomly/Chronic/Diagnosis of ASD/3 weeks + 3 weeks. Five times per week. Before the classroom.	9 children (2 girls)/3–6/5.2/USA	2 groups: EG (n = 9): 15 min of running/jogging followed by a classroom task CG (n = 9): Classroom task not preceded by exercise	AP: Correct academic responses, incorrect, or no academic responses Stereotypic behaviors (hand-and-arm flapping, body rocking, and toe walking) On-task behavior (when seated and producing academic responses) off-task when they did not produce an academic response and were engaged in disruptive behavior (e.g., crying, out of seat, or playing with objects)	Significant improvements were found in correctly responding following exercise (*p* < 0.05). No significant differences were found for on-task behavior or stereotypic behaviors.
Obrusnikova, Bibik, Cavalier, and Manley (2012) [45]	Interventional, cross-over. Contrabalanced/Acute/Diagnosis ASD Asperger/1 day. In the evening.	4 children (all boys)/9–11/USA	EG (n = 4): MVPA with a dog CG (n = 4): MVPA The exercise consisted of 4 fitness stations and 2 skill stations (≈20 min) System for observing fitness instruction time	AP: Academic learning time–physical education	Regardless of the order of the intervention, the MVPA gains (Mgain ¼ 0.62) and the on-task behavior gains (Mgain ¼ 3.52) were larger for the therapy dog condition compared with the peer condition. These findings were confirmed in semistructured individual interviews with the children’s parents and instructors.
Liu and Hamilton (2013) [28]	Interventional, cross-over/Acute/Age, gender, and disorder. Diagnosis ASD/4 days. Four sessions. Before the classroom.	23 children and adolescents (6 girls)/5–13/USA	1 group: Intervention: MVPA for 15 min a day Measures pre and post each season Each day different PA type Heart rate	Children’s behaviors: Stereotypic behavior (was defined as the child not participating in activities in an appropriate manner) and task-engaged behavior (was defined as the child acting appropriately in the current situation while listening to directions, as well as interacting well with others)	Physical activity was identified as moderate or vigorous based on the child’s heart rate. The child was observed for two and a half hours each day and their behaviors were then classified as either stereotypic behavior or task-engaged behavior. No significant behavior differences related to exercise on age, gender, or disorder were observed.
Brand et al. (2015) [35]	Interventional, cross-over/Chronic/Diagnosis of ASD with ICD-10/3-week. Three times per week. In the evening.	10 Children (5 girls)/7–13/10 ± 2.34/Switzerland	1 group: EG: 60 min (30 min bicycle + 30 min of training in coordination and especially in balance) Motor skills (balancing, throwing, some ball skills…)	Sleep-EEG, beginning and at the end of the study	Mild-to-moderate insomnia was reported in 70% of the children. Compared to nights without previous intervention, on nights following intervention sleep efficiency increased (d = 1.07), sleep onset latency shortened (d = 0.38), and wake time after sleep onset decreased for 63% of the sample (d = 1.09), as assessed via sleep-EEG. Mood in the morning, as rated by parents, improved after three weeks (d = 0.90), as did motor skills (ball playing, balance exercise: ds.0.6).
Ketcheson et al. (2016) [33]	Interventional, cross-over/Chronic Diagnosis with ADOS-2 and DSM-IV. Time point with cognitive t score/8 weeks. Five times per week. During the summer.	20 children (5 girls)/4–6/-/USA	2 groups: EG (n = 11): Motor skill instruction for 4 h/day, CG (n = 9): No intervention TGMD-2: Locomotor skills and object control skills)	ADOS-2: Full-scale intelligence quotients MSEL: Cognitive scales, including non-verbal problem solving, fine, receptive language, and expressive language VABS-2: Standardized parental report measure of overall adaptive behavior POPE: Behavior coding system	A significant effect of time (for decreasing minutes) solitary (F(4, 8.76) = 7.94, *p* < 0.01). No significant effects of time (for increasing or decreasing minutes) were found in the remaining POPE-dependent variables, including joint engagement, parallel play, or onlooking. Significant differences between groups in all three motor outcomes, locomotor (F(1, 14) = 10.07, *p* < 0.001, partial η^2^ = 0.42), object control (F(1, 14) = 12.90, *p* < 0.001, partial η^2^ = 0.48), and gross quotient (F(1, 14) = 15.61, *p* < 0.01, partial η^2^ = 0.53) were observed. No comparison between ADOS and MSEL.
Pan et al. (2017) [38]	Interventional, cross-over, Randomly/Chronic/Diagnosis with DSM-IV-TR/2 × 12-weeks. A total of 2 sessions per week (24 sessions). After school.	22 children (all boys)/6–12/9.08 ± 1.75/China	2 groups: 2 TIMES: Time 1: Group A (n = 11): 70 min per session; warm-up (5 min), motor skills (20 min), executive function (20 min), group games (20 min), and cool-down (5 min). Group B (n = 11): No PA session Time 2: Group A: Follow up (No PA session) Group B: 12-weeks intervention Anthropometric measurements and bioelectrical impedance BOT-2: Motor skill proficiency	WCST: Executive function	Both groups of children with autism spectrum disorder exhibited significant improvements in motor skill proficiency (the total motor composite and two motor-area composites) and executive function (three indices of the WCST after 12 weeks of physical activity intervention). In addition, the effectiveness appeared to have been sustained for at least 12 weeks in Group A.
Nakutin and Gutierrez (2019) [20]	Interventional/Chronic/Diagnosis with DSM-V/7-weeks, 2 sessions per week (13 sessions) × 2. During academic activities (9h).	3 children (2 girls)/6–7/-/USA	1 group—2 times: Intervention: 13 sessions. Jogging duration was 12 min, followed by a 5-min cool-down Non-intervention: 13 sessions	BOSS: Academic engagement GNG: Inhibition Digits forward and backward tasks: Working memory STP: Social validity	Large effect sizes for academic engagement were found. No significant improvement in working memory and inhibition was observed.
Tse et al. (2019) [31]	Interventional, Randomly/Chronic/Diagnosis with DSM-V/12-weeks, 2 sessions per week (24 sessions). Before the classroom.	40 children (8 girls)/8–12/9.95 ± 1.17/China	2 groups: EG (n = 19): 45 min per session. Warmup (10 min), basketball (30 min), and cool-down (5 min). CG (n = 21): No intervention 4 sleep parameters (ActiGraph GT3X)	GNG: Inhibition control CBTT, FDS, BDS: Working memory	Significant improvement in inhibitory control (EG) was found. No significant improvement in working memory (all) was observed.
Andy (2020) [46]	Interventional, cross-over, Randomly/Chronic/Diagnosis with DSM-V/12-weeks, 4 sessions per week (48 sessions). At school.	27 children (4 girls)/8–12/10.07 ± 1.10/China	2 groups: EG (n = 15): 30 min per session, jogging. Moderate to vigorous (heart rate monitor) CG (n = 12): No intervention	ERC: Emotion regulation CBCL: Behavioral functioning	Significant improvement in emotion regulation and a reduction in behavioral problems were observed.
Zhang et al. (2020) [47]	Interventional/Chronic/Diagnosis with DSM-V/12-weeks, five days per week (60 sessions). At school.	33 children (5 girls)/3–6/4.92 ± 0.67/China	2 groups: EG (n = 18): 40 min/session, mini-basketball training program. Moderate intensity (129–149 heart beats per minute). CG (n = 15): No intervention	CHEXI: Executive functions SRS-2 and RSR-R: Core symptoms (social communication impairment and repetitive behavior)	Significantly better performances in working memory (*p* < 0.01) and regulation behavior (*p* < 0.05) were observed.
Rafiei et al. (2021) [48]	Interventional/Chronic/Diagnosis with DSM-V/8-weeks, three times per week, 35 min/session (24 sessions). After school.	60 children (3 girls)/6–10/8.45 ± 1.43/Iran	3 groups: EG1 (n = 20): SPARK EG2 (n = 20): Kinect CG (n = 20): No intervention MABC-2: Motor skills	WCST: Executive function	Significantly better performances in motor skills (aiming and catching) (*p* < 0.05) on EG1 were observed. EG2 showed more correct responses than the EG1 and CG (conceptual responses and perseverative errors) (*p* < 0.01).
Golden et al. (2022) [30]	Interventional/Acute/Diagnosis with DSM-V/1 session for each condition (20 min). After school.	8 boys/8–11/10.60 ± 1.52/USA	3 conditions: EG1 (n = 8): Active video gaming EG2 (n = 8): Brisk walking CG (n = 8): Sedentary video gaming	Flanker task: Response time and accuracy	A significantly higher percentage on incongruent in reaction time for the CG was observed. EG1 and EG2 improved in accuracy. EG2 participants improved significantly in reaction time in the congruent condition.
Haghighi et al. (2022) [49]	Interventional/Chronic/Diagnosis with DSM-V/8 weeks, three times per week, 40–60 min/session/after school.	16 children (7 girls)/6–10/9.00 ± 1.31/Spain	2 groups: EG (n = 8): Combined physical training (ball game, rhythmic movements, and resistance training) CG (n = 8): No intervention MABC-2: Motor skills Physical fitness (cardiorespiratory fitness, muscle strength, flexibility, balance, and agility)	GARS-2: Stereotypic behaviors, communication, and social interaction	Significantly better performances in indicators of social skills such as stereotypic behavior and communication on EG were observed. Significantly better performances in handgrip strength, upper and lower body power, flexibility, balance, and agility were observed (*p* < 0.05).
Marzouki et al. (2022) [50]	Interventional/Chronic/Diagnosis with DSM-V/8 weeks, two times per week, 50 min/session (16 sessions/after school.	28 children (7 girls)/6–7/6.30 ± 0.50/Germany	3 groups: EG1 (n = 10): Technical aquatic activities program EG2 (n = 10): Game-based aquatic activities program CG (n = 8): No intervention TGMD-2: Motor skills	GARS-2: Stereotypic behaviors, communication, and social interaction ERC: Emotional regulation	Both forms of swimming had a positive effect on gross motor skills and stereotypic behaviors. Changes in emotional functioning, with respect to the control group, were observed (*p* < 0.05).
Ludyga et al. (2023) [51]	Interventional/Acute/Diagnosis with DSM-V/1 session/20 min/after school.	29 children (1 girl)/7–12/10 ± 2/Swiss	2 condition: EC (n = 8): Acute aerobic exercise (20-min moderately intense cycling) CC (n = 8): No intervention	Face recognition task EEG recordings + Eyetracking: Reaction time, correct responses, and accuracy rates	A greater increase in reaction time in the exercise groupcompared to the CC was observed, as well as impaired face recognition following aerobic exercise.

Note: ADOS = Autism Diagnostic Observation Schedule; AP = Academic Performance; ASD = Autism Spectrum Disorder; BDS = backward digit span test; BOSS = Behavioral Observation of Students in Schools; BOT = Bruininks–Oseretsky Test; CBCL = Child Behavior Checklist; CBTT = Corsi block tapping task; CCTT = Children’s Color Trails Test; CG = Control Group; CHEXI = Childhood Executive Functioning Inventory; DSM = Diagnostic and Statistical Manual of Mental Disorders; EEG = Electromyography; EG = Experimental Group; ERC: Emotion Regulation Checklist; GARS-2 = Gilliam Autism Rating Scale, second edition; GNG = go/no-go task; ICD = International Classification of Diseases; MABC-2 = Movement Assessment Battery for Children, second edition; MSEL = Mullen Scales of Early Learning; MVPA = Moderate to vigorous Physical Activity; PA = Physical Activity; POPE = Playground Observation of Peer Engagement; SPARK = Sports, Play, and Active Recreation for Kids; SRS-2/SRS-R = Social Responsiveness Scale, second edition/revised; STAI = State-Trait Anxiety Inventory; TGMD = Test of Gross Motor Development; Family Adaptation and Cohesion Evaluation Scales; TSP: Scale of Treatment Perceptions; VABS = Vineland Adaptive Behavior Scale; WCST = Wisconsin Card Sorting Test.

**Table 4 behavsci-14-00330-t004:** A summary of practical proposals of physical activity to obtain effects on cognition, behavior, and motor skills in youth with autism spectrum disorder.

Stimulus	Age	Application			Effects		
		Continuity of the Stimulus	Intensity	Type/Moment of Day	Cognitive and Academic Performance	Behavior	Motor Skills
15–30 min aerobic exercise (jogging, bicycle, or swimming)	3–12	5 days/week	50–70% MHR	Before classroom	>Attention >Academic engagement <Incorrect responding >Correct responding	>Activity ≈Behaviors	≈Stereotypic behaviors
15–20 min physical exercise (coordination and aerobic), exergames (AURASMA App)	6–18	5 days/week	>75% MHR	Activity break lessons/Classroom/In the evening	>Executive function	>Behaviors	<Stereotypic behaviors <Repetitive behaviors
20–40 min circuit with animal therapy (horses or dog)	6–12	1–2 days/week	MVPA	In the evening	>Task-engaged behavior	>Behaviors	≈
30–60 min coordination and balance training	6–12	3–5 days/week	MVPA	In the evening/Homework break	>Neural activity	≈	>Coordination and balance
40–75 min aerobic exercise or sport (basketball, tennis, or football)	6–18	2–3 day/week	MVPA	In the evening/Homework break	>Executive function	<Stress and anxiety >Relaxation	Total motor composite

Note: A = Acute effect; C = Chronic effect; MHR = Maximal heart rate; MVPA = Moderate vigorous physical activity; ≈ = No consensus; > = Higher/Increase; < = Less/Decrease.

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
