# Peer review of "Effects of Physical Activity on Cognition, Behavior, and Motor Skills in Youth with Autism Spectrum Disorder: A Systematic Review of Intervention Studies"

_behavsci, 2024, doi:10.3390/bs14040330_

Round 1

Reviewer 1 Report

Comments and Suggestions for Authors

Due to the title of the paper, I liked to read the paper to get better understanding of the topic. The following issues popped up which can improve the quality of the paper.

1.    The main note: as a reader, I like to see if there is a conclusion in the number of sessions/week, duration/session, intensity, and type of activity. This would help in practice or future research.

a.    Table 4’s title claims about recommended physical activities. However, it is more what they have done not what is recommended based on the studies.

2.     There are studies with under 10 participants. This would make those studies preliminary studies and hardly can be used a systematic review. (Table 3). These should be single subject studies.

3.     One study is over a year which makes its results hard to justify since the effect of other factors can play a role in the results.

4.     How many studies had follow-up?

5.     There are citations which do not look correct for the paper. For instance, the following paper does not talk about autism at all.

Hermassi, S., Chelly, M. S., Michalsik, L. B., Sanal, N. E., D. Hayes, L., & Cadenas-Sanchez, C. (2021). Relationship between fatness, physical fitness, and academic performance in normal weight and overweight schoolchild handball players in Qatar 448 State. Plos one, 16(2), e0246476. https://doi.org/10.1371/journal.pone.0246476

6.     The authors ranked papers based on 6 factors which is interesting. I liked to see other factors such as the number of subjects.

7.     There are similar systematic reviews published in the same period that you have searched. Why didn’t you include them?

Effects of physical activity and exercise-based interventions in young adults with autism spectrum disorder: A systematic review - Vaishnavi Shahane, Amanda Kilyk, Sudha M Srinivasan, 2024 (sagepub.com)

8.     Writing issues

a.     What is “height sessions of tri-cycling”?

b.    What is meant by “Brain activity” is Table 4.

c.      “Upported” should be supported in table 3

d.    “over time, our results highlight the potential of exercise and relaxation for improving symptoms of stress”. If this is a sentence from the original paper, it should be put in quotation.  

e.      In Table 4, the first row’s stimulus is name MVPA while it is not clear what kind of activity it is. 

Comments on the Quality of English Language

I mentioned some issues with the writing above. Please check them. There are other issues to be fixed.

Author Response

we attach the reply to reviewer 1

Reviewer 2 Report

Comments and Suggestions for Authors

• Review minor spelling errors (such as the lowercase letter after a colon in the title) and grammar throughout the document. • There are errors in some APA format references. For example:

Bombonato, C., Del Lucchese, B., Ruffini, C., Di Lieto, M. C., Brovedani, P., Sgandurra, G., Cioni, G. & Pecini, C. (2023). Far 413 Transfer Effects of Trainings on Executive Functions in Neurodevelopmental Disorders: A Systematic Review and Metanalysis. 414 Neuropsychology Review, 1-36. https://doi.org/10.1007/s11065-022-09574-z

Cibralic, S., Kohlhoff, J., Wallace, N., McMahon, C., & Eapen, V. (2023). Emotional Regulation and Language in Young Children 419 With and Without Autism Traits. Journal of Early Intervention, 0(0), https://doi.org/10.1177/10538151231176188

Haghighi, A. H., Broughani, S., Askari, R., Shahrabadi, H., Souza, D., & Gentil, P. (2022). Combined physical training strategies 441 improve physical fitness, behavior, and social skills of autistic children. Journal of Autism and Developmental Disorders, 1-9. 442 https://doi.org/10.3390/biology11050657

• Are the keywords appropriate? And in that order?

• In the introduction:

o The diagnostic criteria of the current version of DSM-5-TR should be followed.

o There are no subtypes in ASD. It is all Autism Spectrum Disorder.

o How are cognition, behavior, and motor skills affected? A brief overview is provided in one paragraph. Lack of depth. o What about patterns of communication and social interaction?

• In the method:

o IT IS STATED: "population of students between three and 18 years," but there are articles with a sample of 21 years (Anderson-Hanley et al., 2011), 27 (Hillier et al, 2011). Is this correct?

• Results:

o Should we outline table 3?

o Tables complicate understanding: Would it be better to include them in the annexes? And a summary in the text?

• Conclusions are very poor.

o Lack of depth in the conclusions: could have delved into the findings related to the stated objectives. In other words, based on the benefits found at the cognitive, behavioral, and motor skills levels, what would be the foundations for the development of the didactic application?

o Age differences?

o Gender differences?

o It is not stated how this article contributes. In other words: what are the future lines of research? Would new lines emerge from the study?

Author Response

we attach the reply to reviewer 2

Reviewer 3 Report

Comments and Suggestions for Authors

Very interesting topic.

Please edit throughout for grammar, punctuation, APA, etc.  Sentences should not begin with numbers - the numbers should be spelled out. Use appropriate APA for listing items in parentheses (line 61).

A number of incomplete thought is this paper (line 68).

Paragraph that begins on line 80 is only two sentences -- please expand.

Introduction sentence seems to be very appropriate.

Table 1 is very hard to decipher.

List search words used in text of paper. 

Paragraph under Selection Criteria is not formatted correctly.

Paragraph that starts on line 150 is not formatted correctly.

Not sure how Table 2 adds to this paper.  Maybe put as an Appendix. 

Table 3 is very good, but a short summary of all of the articles included in this study should be in the text.

Section 3.2 of text is unclear. It starts off with one of four studies then moves to five studies. Maybe add tables for Sections 3.2 and 3.3 to highlight specific results. 

Not sure Table 4 is needed.  Isn't that information already presented previously?

Nice review of limitations of the study. 

Comments on the Quality of English Language

Major edits are needed, but overall it is readable.

Author Response

we attach the reply to reviewer 3

Round 2

Reviewer 1 Report

Comments and Suggestions for Authors

Table 4 is not a set of recommendations. It is rather a summary of what the articles have done. 

Comments on the Quality of English Language

Still the writing needs to be improved.

Author Response

Thank you so much for your suggestion. We have changed the title of the table 4. The final title is: 

Table 4. Summary of practical proposals of physical activity to obtain effects on cognition, behavior, and motor skills in youth with autism spectrum disorder.

Kind regard,

Reviewer 2 Report

Comments and Suggestions for Authors

Thank you for the changes made. The article has improved significantly with the additions.

Author Response

Thank you so much for your kindly response.

Kind regards

Reviewer 3 Report

Comments and Suggestions for Authors

Thanks for addressing reviewer comments. Manuscript looks much better.

Author Response

Thank you so much for your kind response. 

Kind regards,